# GPR168 functions as a tumor suppressor in mouse melanoma by restraining Akt signaling pathway

Xiang Guo[1,2], Zongliang Guo[3], Peirong Bai[1,2], Congfang Guo[4], Xuewei Liu[5], Kaiyi Zhu[1,2], Xiaoyan Li[6]*, Yiyan Zhao[1,2]*

1 Shanxi Academy of Medical Sciences, Shanxi Bethune Hospital, Tongji Shanxi Hospital, Third Hospital of Shanxi Medical University, Taiyuan, China, 2 Tongji Medical College, Tongji Hospital, Huazhong University of Science and Technology, Wuhan, China, 3 Department of General Surgery, Chinese Academy of Medical Sciences, Shanxi Province Cancer Hospital, Shanxi Hospital Affiliated to Cancer Hospital, Cancer Hospital Affiliated to Shanxi Medical University, Taiyuan, China, 4 Tianjin First Central Hospital, Tianjin, China, 5 College of Veterinary Medicine, Institute of Animal Biotechnology, Shanxi Agricultural University, Taigu, Jinzhong, China, 6 Department of Blood Transfusion, Shanxi Provincial People's Hospital, Affiliate of Shanxi Medical University Taiyuan, Taiyuan, Shanxi, China

* xiaoyanli5959@163.com (XL); 490051225@qq.com (YZ)

**Data Availability Statement:** All relevant data are within the manuscript and its Supporting Information files.

**Funding:** This study was supported by grants from the Fundamental Research Program of Shanxi

## Abstract

Malignant melanoma (MM) is a malignant tumor associated with high mortality rates and propensity for metastasis. Despite advancement in treatment, the incidence of MM continue to rise globally. GPR168, also known as MrgprF, is a MAS related GPR family member. The low expression of GPR168 has also been reported in many malignant tumors including MM. In the study, the statistical analysis from The Cancer Genome Atlas (TCGA) revealed a significant down regulation of GPR168 in melanoma compared to normal melanocytes, underscoring its importance in MM. The aim of the present study is to investigate the affect of GPR168 overexpression and elucidate its molecular mechanisms in MM cells. In addition, we used mouse melanoma B16-F10 cell line and xenograph tumor model to explore the function of GPR168 in melanoma. Our findings demonstrate that GPR168 overexpression could inhibit B16-F10 cell proliferation, migration, and xenografts tumor growth. Further, mechanistic studies revealed that GPR168 affected B16-F10 progress through Akt signal pathway with the decreased expression of p-Akt, p-GSK-3β, β-catenin, Myc, CyclinD1 and CDK4. In order to validate these findings, a rescue experiment was formulated employing GPR168 polyclonal antibody (Anti-GPR168 pAbs) to block GPR168 functionality. The addition of Anti-GPR168 pAbs into the culture medium restored both cell proliferation and migration. In conclusion, the overexpression of GPR168 in mouse melanoma B16-F10 cells suppressed proliferation and migration through the Akt signaling pathway. These findings collectively propose GPR168 as a promising novel tumor suppressor in MM, suggesting its potential as a therapeutic target in future interventions.

Province [2022203021222348] to XG, Scientific research start-up fund project for talent introduction of Shanxi Bethune Hospital [2023RC22, 2023RC41] to XG and YZ respectively, and the Open Fund Project of Shanxi Key Laboratory of Nucleic Acid Biopesticides to YZ.

**Competing interests:** The authors have declared that no competing interests exist.

**Abbreviations:** MM, malignant melanoma; TCGA, The Cancer Genome Atlas; RTKs, receptor tyrosine kinases; GPCRs, G protein coupled receptors; PIP2, phosphatidylinositol-(3,4)-P2; PIP3, phosphatidylinositol-(3,4,5) P3; CVAA, Cross-Value Association Analysis; DEG, Differentially Expressed Genes; MHC, Major histocompatibility complex; RT-qPCR, Quantitative real-time PCR; IHC, Immunohistochemical; BrdU, bromodeoxyuridine; H&E, Hematoxylin and eosin staining; CCK-8, Cell Counting Kit8; GSK-3β, Glycogen synthase kinase 3 beta; CDK4, cyclin dependent kinase 4; IPTG, Isopropyl-beta-D-thiogalactoside; Akt, phosphoinositide kinase pathway.

## Introduction

Malignant melanoma (MM) is a type of skin cancer caused by the abnormal proliferation of melanocytes. Over the recent years, there has been a notable global surge in the incidence of MM [1, 2]. Advanced age and Caucasian race are the two main factors of MM according to epidemiological data from United States. Additionally, tanning has also been indicated as a risk factor [3]. Although melanoma accounts for about 1% of all skin cancers, it caused 90% of skin cancer deaths due to its high invasiveness [4]. MM has the characteristics of rapid proliferation, anti-apoptosis, unlimited replication, and significant increase in melanin content [5]. Early detection of melanoma is crucial *in situ*, because it is "curable" at early stage and the treatment of this stage is mostly surgery, but it is difficult to treat after metastasis [6]. The 5-year survival rate for patients after early surgery is 95%, while the median survival time for advanced patients after surgery is only 2 to 8 months, and the 5-year survival rate is less than 5% [6–9]. With the in-depth development of MM research in the past decades, the prognosis of MM patients has been greatly improved, but its high invasiveness and metastasis are the main causes leading to treatment failure [4].

The mechanism of MM was found to be complex, involving epithelial-mesenchymal transition and angiogenesis in the tumor microenvironment, invasion of melanoma cells and degradation of extracellular matrix [9, 10]. Advanced MM mainly involves lymph node and brain metastases, which determine the staging, treatment options and prognosis evaluation of MM [11, 12]. In addition, cumulative alternations in multiple genes and signaling pathways are also the main reason. Most MM have potential mutations occurring mainly in components of the phosphoinositide kinase (PI3K/Akt) signaling pathways and mitogen-activated protein kinase (MAPK) [6, 13]. The AKT pathway is a signaling pathway involved in phosphatidylinositol, which is activated by receptor tyrosine kinases (RTKs) and G protein coupled receptors (GPCRs), leading to the increased conversion of phosphatidylinositol-(3,4)-P2 (PIP2) to phosphatidylinositol-(3,4,5) P3 (PIP3), as well as high level of phosphorylation on Akt proteins [6]. The MAPK pathway is composed of three protein kinases, MAP3K-MAP2K-MAPK, which transmit upstream signals to downstream response molecules through sequential phosphorylation [13]. Both of these signaling pathways are related to the proliferation, migration and carcinogenesis of MM [13–16]. However, there is still much work remaining to be done to improve the earlier detection of lethal MM, due to lack of reliable diagnostic biomarkers.

With the increase of MM patients, an in-depth research on melanoma has become an urgent need for medical workers around the world [4]. Through literature exploration, numerous pan-cancer biomarkers have been identified, showing consistent dysregulation across various forms of human cancer [17]. By using comprehensive bioinformatic analysis, combined with MM related datasets available from other web-sources, we identified that GPR168, a member of GPR family related to MAS, was decreased in MM. Therefore, we decided to explore the function of GPR168 in the B16-F10 mouse melanoma cell line, elucidate the molecular mechanism, and propose a potential therapeutic approach for MM.

## Materials and methods

### Construction of plasmids

The CDS region of *GPR168* was obtained by PCR method with the primers:
    F-5'ATGTGTCCTGGTATGAGCGAG3',
    R-5'TCAGGATGCGTTCCCAGAGG3'.

The PCR product was cloned into the vector pCDH-CMV-MCS-EF1-copGFP-T2A-Puro (pCDH-Vec) to construct the plasmid pCDH-CMV-MCS-EF1-copGFP-T2A-Puro-GPR168 (pCDH-GPR168), and it was further verified by sequencing.

## Cell culture and transfection

293T and B16-F10 cell lines (stored at Shanxi Bethune Hospital, Shanxi, China) were cultured at 37°C in a 5% $CO_2$ atmosphere. 293T were cultured in DMEM-High glucose medium (Hyclone, America) and B16-F10 cell lines were cultured in RPMI-1640 Modified medium (Hyclone, America). These cell lines were cultured in medium supplemented with 10% fetal bovine serum and 1% penicillin-streptomycin. Following the manufacturer protocol (BGI Beijing China), virus was generated from 293T cells and B16-F10 cells were infected twice with viral supernatants containing 4 μg/ml polybrene at 48 h and 72 h, respectively.

## Quantitative real-time PCR (RT-qPCR)

mRNA expression was determined by quantitative real-time PCR (RT-qPCR). Total RNA from cells was isolated by TRizol RNA isolation reagents (Takara, Beijing, China) and one microgram of the total RNA was used to synthesize the first-strand cDNA using TaqMan reverse transcriptase kit (Takara Bio, Beijing, China). One microgram cDNA was subjected to RT-qPCR using FastStart Universal SYBR Green Master Mix (Roche, Switzerland) to detect the relative expression. Reactions were performed in triplicate. *β-actin* was used as internal control for normalization among cell samples. All the gene specific-primers used for RT-qPCR are listed in Table 1.

## Western blotting

Cells were harvested and prepared in the Cell Lysis Buffer (Beyotime, Shanghai, China) and the protein concentration was measured by Bio-Rad Protein Assay Kit (Bio-Rad, CA, USA). Then the supernatants were boiled at 95°C for 15 min, which were separated by electrophoresis and transferred to polyvinylidene fluoride membrane (Millipore). Primary antibodies used in this study were as follows: β-actin (CWBIO, 1:5000), Akt (Proteintech, 1:1000), p-Akt (Proteintech, 1:1000), GSK-3β (Proteintech, 1:1000), p-GSK-3β (Proteintech, 1:1000), CyclinD1 (Abcam, 1:1000), Myc (Proteintech, 1:1000), β-catenin (Proteintech, 1:1000), CDK4 (Proteintech, 1:1000). The secondary antibodies conjugated with horseradish peroxidase goat anti-mouse IgG (CWBIO, 1:20000) or horseradish peroxidase-cojugated goat anti-rabbit IgG (CWBIO, 1:20000). After washing six times for 10 min each with TBST buffer, the membranes

**Table 1. Sequences for the primers used in RT-qPCR.**

| Gene | Sequence (5′–3′) | Application |
|---|---|---|
| *GPR168*-F | GTGTCCTGGTATGAGCGAGG | RT-qPCR |
| *GPR168*-R | AGGGGTCCTCTTGATGGAGA | RT-qPCR |
| *β-actin*-F | TTGCTGACAGGATGCAGAAG | RT-qPCR |
| *β-actin*-R | ACATCTGCTGGAAGGTGGAC | RT-qPCR |
| *Cyclin D1*-F | TGTCTTACCACCGCCTCAC | RT-qPCR |
| *Cyclin D1*-R | CTCCTCTTCCTCCTCCTCCT | RT-qPCR |
| *CMY*-F | ATCACAGCCCTCACTCAC | RT-qPCR |
| *CMY*-R | ACAGATTCCACAAGGTGC | RT-qPCR |
| *CDK4*-F | CAGTTTCTAAGCGGCCTGGA | RT-qPCR |
| *CDK4*-R | TCCTCCTTGTGCAGGTAGGA | RT-qPCR |

were incubated at 37˚C for 2 h with horseradish peroxidase conjugated secondary antibodies raised against rabbit or mouse IgG, and followed by analyzing with the MiniChemi imaging systems (Beijing, China).

## Cell proliferation and migration assays

The proliferation ability of B16-F10 cells was evaluated by Cell Counting Kit-8 (CCK-8) and growth curve assays [18, 19]. The 1000/well cells were planted in a 96-well cell culture plate. After cell adhesion, 10 μL of CCK-8 (BBI, Shanghai, China) reagent was added to each well, and OD values were detected at 450 nm at 0 h, 3 h, 6 h, 9 h, 12 h and 15 h, respectively. In the growth curve assay, 2000 cells were planted in each well of the 24-well plate [20, 21]. The cells from each well were then trypsinized and determined using automatic cell analysis counter, which was repeated three times a day for each group.

The migration ability of B16-F10 cells was evaluated by transwell and wound healing assays [22]. For transwell assay, cells in serum-free medium were planted on the uncoated insets and incubated using 24-well chemotaxis chambers (Corning cell culture inserts, 8 μm pore size). Medium supplemented with 10% fetal bovine serum acted as chemo-attractant, which was added to the bottom wells. After 24 hours of incubation, removed the non-migrating cells and stained the migrating cells to the lower surface with cresyl violet (Sigma, USA). Stained cells in the entire fields were counted under microscope. For wound healing assay, cells were planted in a six-well plates at $1\times10^6$ cells/well in 2 mL of culture medium [20, 22]. After 12 h, a wound was scratched with the tip on the adherent cell monolayers and imaging was performed at 0 h, 12 h, 24 h and 48 h at 8–12 positions along each well.

## Immunofluorescence staining

Cells were fixed with 4% paraformaldehyde for 20 min, permeabilized with 0.1% Triton X-100 PBS and blocked by 10% goat serum. The cells were incubated by primary antibody Ki67 (Bioss, 1:100) at 4˚C overnight and probed with fluorescence-conjugated secondary antibodies (Bioss, 1:200) at room temperature for 2 h. Finally, the cells nuclei were stained with DAPI (Beyotime, China). Five randomly selected fields/samples were imaged on a fluorescence microscope and quantified.

## Xenograph tumor model in nude mice

A total of 10 male 6-week-old nude mice were randomly divided into two groups to construct the tumor xenografts of MM. The mice of control group were injected with $1x10^6$ pCDH-Vec B16-F10 cells, while the mice of experimental group were injected with the same amounts of B16-F10 cells with GPR168 overexpression (see method 2.2). Tumor sizes of mice were measured every 3 days for 30 days using caliper. The tumor volume was calculated by the following formula: tumor volume = (length) × (width)$^2$ × 0.5. When a tumor reached ethical limits, all mice were killed, and the tumors were collected and weighed.

## Immunohistochemical staining

Xenograft tumors were collected and fixed with 4% formaldehyde overnight, sectioned at 5 μm after paraffin-embedded, deparaffinized by xylene, rehydrated with gradient ethanol, and subjected to antigen retrieval. Antigen retrieval was performed by placing the slides at 95˚C for 20 min in a microwave oven and allowed to cool for 1 h at room temperature. The slides were again washed three times with TBS, and nonspecific binding was blocked by pre-incubation with 5% BSA for 30 min at room temperature. Slides were then incubated for 2 h at

4˚C with primary antibody Ki67 (Bioss, 1:100) in the blocking buffer. After washing the slides three times with TBS, sections were subsequently treated with HRP-labeled second antibody (CWBIO, 1:200) for 40 min. Diaminobenzidine was used as a chromogen followed by slight hematoxylin counterstaining. The slides were then dehydrated, cleared with xylene and mounted with dibutyl phthalate xylene. The positive signals were imaged under microscope.

### Institutional review board statement

The study was conducted according to the guidelines of the Declaration of Helsinki, and approved by the Ethics Committee of Shanxi Medical University (2017(050)).

### Statistical analyses

Data were presented as mean ± SEM. $p$ values were calculated by either unpaired or paired two-tailed Student's $t$ test, *$P<0.05$, **$P<0.01$, and ***$P<0.001$. All analyses were performed using GraphPad Prism software. All the experiments were performed at least three times.

## Results

### GPR168 was weakly expressed in melanoma

We applied Cross-Value Association Analysis (CVAA) to large-scale pan-cancer transcriptome data generated by The Cancer Genome Atlas (TCGA). [17] Numerous new Differentially Expressed Genes (DEGs) have been discovered, and GPR168 is one of them. To investigate the potential roles of GPR168 in tumorigenesis, we examined the expression profile of GPR168 in human tissues using RNA sequencing data retrieved from the GTEX database (http://www.gtexportal.org), and showed that *GPR168* is ubiquitously expressed in most human tissues. At same times we found that *GPR168* expression is diminished in majority of cancer types when compared to their respective normal tissues, except GBM (glioblastoma) and head and neck squamous cell carcinoma (HNSC) (Fig 1A). Furthermore, we investigated the correlation between GPR168 and clinicopathological features of MM. The GPR168 expression between melanoma and normal tissues samples was analyzed using microarray data sets from the TCGA cohorts downloaded from CEPIA (http://gepia.cancer-pku.cn/). In cohorts, the expression of GPR168 was significantly lower in melanoma compared to normal skin tissues (Fig 1B). Then, cBioPortal web (1089 melanoma patients) was used to analyze the relationship between *GPR168* gene and melanoma patients (https://www.cbioportal.org/). The results showed that the Diploid accounts for 57.2%, the Shallow Deletion accounts for 27.1% and the Gaina accounts for 10.7% (Fig 1C). All the statistical data showed that GPR168 played an important role in MM. Subsequently, we used the B16-F10 cell line *in vitro* and xenograph tumor model *in vivo* to explore the function of GPR168. Analysis of mRNA and protein expression levels of GPR168 in B16-F10, compared with mouse melanocytes cells, revealed a similarly low expression of GPR168 in B16-F10 (Figs 1D–1F and **S1**).

### Overexpression of GPR168 inhibited B16-F10 proliferation and migration

We then enhanced *GPR168* mRNA expression with independent lenti-viral, and validated the overexpression efficiency by comparing relative mRNA expression levels in pCDH-GPR168-targeted B16-F10 cell lines with that in scramble pCDH-Vec control cells. The RT-PCR and western blot results demonstrated that the B16-F10 cell line with stable GPR168 expression was successfully constructed (Figs 2A–2C and **S2**). As expected, the CCK-8 assay results showed that GPR168 overexpression inhibited the proliferation of B16-F10 cells by 35% compared to the control group (pCDH-Vec) (Fig 2F). Growth curve assay showed that

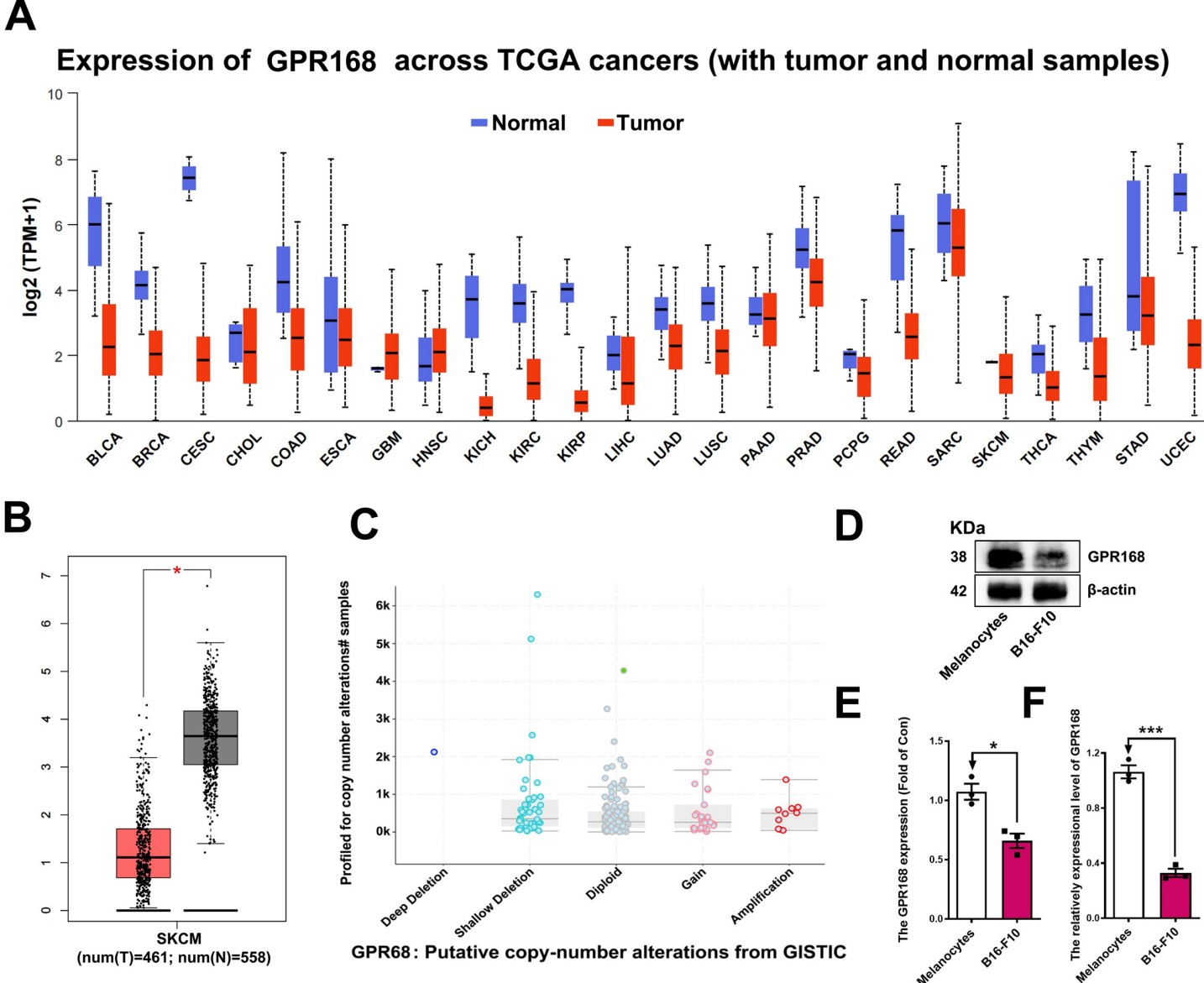

**Fig 1. GPR168 was weakly expressed in melanoma.** (A) GPR168 is lowly expressed in most cancers except GBM (glioblastoma) and head and neck squamous cell carcinoma (HNSC). (B) The expression of GPR168 was lower in melanoma compared to normal tissues, and the microarray data sets from The TCGA cohorts was downloaded from CEPIA. (C) Putative copy-number alterations of 1089 melanoma patients from Genomic Identification of Significant Targets in Cancer (GISTIC), related database was indicated from web cBioPortal (cBioPortal for Cancer Genomics). (D-F) Western blotting and RT-qPCR results showed the protein expression (D, E) and the relative *GPR168* mRNA (F) in mice melanoma cell line B16-F10, compared to normal melanocytes cells (normalized to *ß-actin*), respectively. *$P < 0.05$, **$P < 0.01$, ***$P < 0.001$, $t$-test.

pCDH-GPR168 reduced the proliferation of B16-F10 cells by 40% compared to control on day six (Fig 2G). The mutual verification of these results showed that stable overexpression of GPR168 inhibited the proliferation of B16-F10 cell *in vitro*. Furthermore, we investigated the impact of GPR168 on cell proliferation by affecting the cell cycle. Ki67, an antigen related to proliferating cells, implicates in mitosis and is essential in cell proliferation, which is used clinically to label proliferating cells [23, 24]. The Immunofluorescence results for Ki67 further confirmed that GPR168 overexpression influenced the reduced expression of Ki67 (Fig 2D). The statistical results showed that the proportion of Ki67 positive in the control group was 38%,

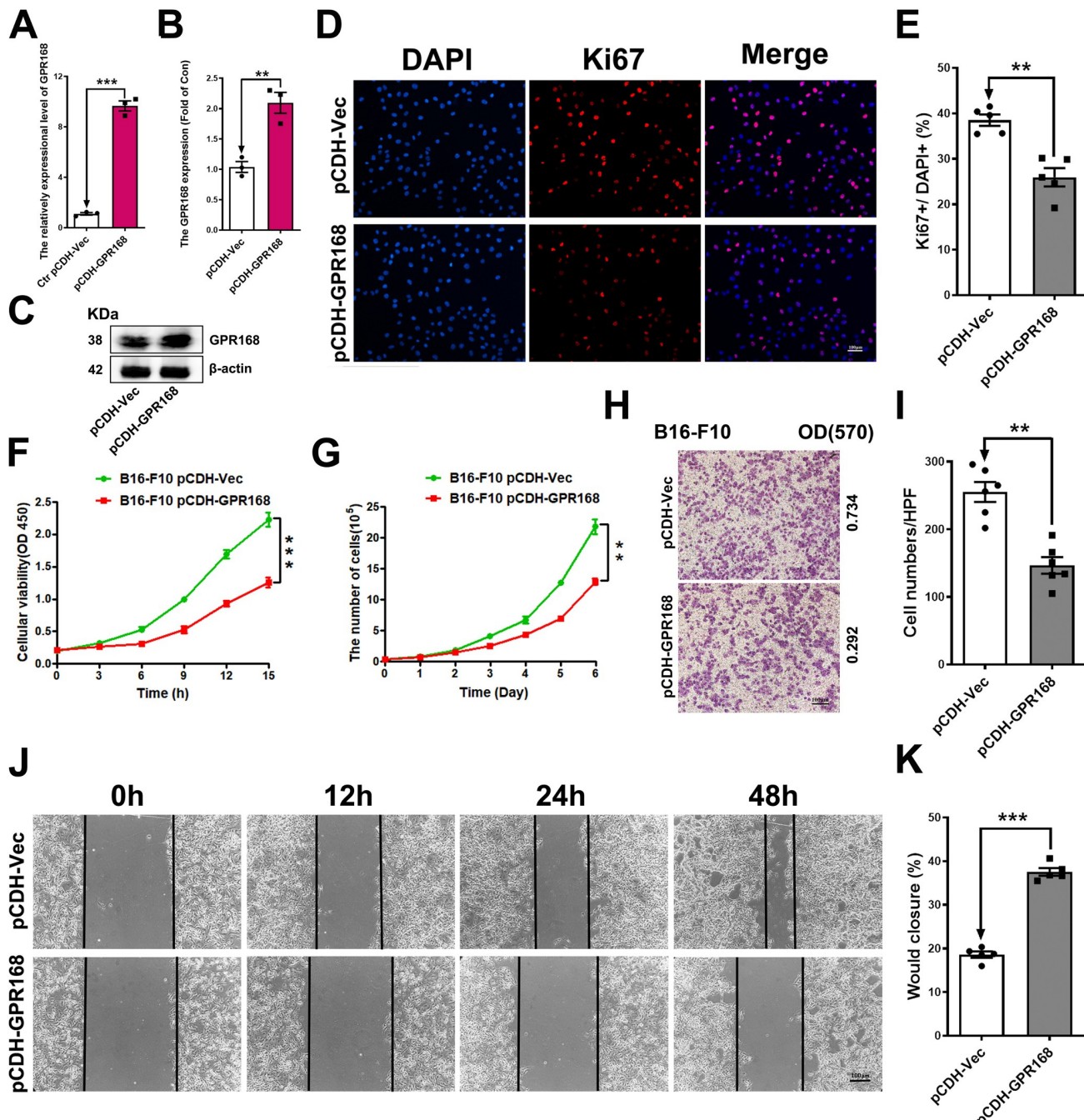

**Fig 2. Overexpression of GPR168 inhibited B16-F10 cell proliferation and migration.** (A-C) The efficiency of pCDH-GPR168 (GPR168 overexpression) was verified by RT-qPCR (A) and western blotting (B, C), compared to scramble pCDH-Vec. (D, E) GPR168 overexpression inhibited cell proliferation in B16-F10 cells by Ki67 Immunofluorescence assay. Scale bar is 100 μm. (F, G) Result of CCK-8 assay (F) and Growth Curve (G) showed that overexpression of GPR168 significantly inhibited B16-F10 cell proliferation. (H, I) overexpression of GPR168 decreased the migration ability of B16-F10 by transwell assay at OD570. Scale bar is 100 μm. (J, K) Wound healing assay showed that overexpression of GPR168 inhibited the migration of B16-F10 cells. Scale bar is 100 μm. *$P < 0.05$, **$P < 0.01$, ***$P < 0.001$, *t*-test.

compared to 26% in the experimental group with a significant difference ($P$ = 0.0004) (Fig 2E), suggesting that GPR168 overexpression inhibited the proliferation of B16-F10 cells.

The transwell assay and wound healing assay were used to perform the migration test. The results showed that GPR168 overexpression inhibited the migration of B16-F10 cells. The statistical results of the number of cells passing through the chamber showed significant differences between the two groups ($P$ = 0.007) (Fig 2H and 2I). Wound healing assay also showed the similar results. The statistical analysis showed that the wound was closed by 38% at 48 h by pCDH-GPR168, compared to the wound closure of 19% by pCDH-Vec ($P$<0.0001) (Fig 2J and 2K). All the results showed GPR168 overexpression in B16-F10 cells dramatically decreased the cell migration.

## Overexpression of GPR168 inhibited the growth of xenograft tumors

To assess the function of GPR168 in MM progression *in vivo*, the xenograph tumor model was constructed. After 30 days, tumor tissues were collected. The results showed that in the pCDH-GPR168 group, the size, volume and weight of melanoma were smaller compared to those in the control group (Fig 3A–3C). The Ki67 IHC in the MM tumor tissues showed significant difference while the H&E staining results showed no significant difference between the experimental group and the control group. These findings displayed that the GPR168 overexpression inhibited cell proliferation *in vivo* (Fig 3D). The statistical results showed that the proportion of Ki67 positive in the experimental group was 21% and that in the control group was 35%, with a statistically significant difference between the two groups ($P$ = 0.0071) (Fig 3E). Taken together, our findings demonstrate that GPR168 overexpression in B16-F10 melanoma cells led to the suppression of tumor growth in nude mice.

## GPR168 functions through Akt signaling pathway

The cell experiments *in vitro* and xenograph tumor model *in vivo* showed that overexpressed CPR168 in B16-F10 cells inhibited the proliferation and migration. Then we wanted to explore GPR168 mechanism in B16-F10 melanoma cells. It is reported that MM is mainly caused by MAPK and PI3K/Akt signaling pathways [4]. MAPK signaling pathway is involved in regulating many important cellular physiological processes of tumor, such as cell growth, differentiation and adaptation to environmental stress [16, 25]. The AKT signaling pathway associated with proliferation, migration and carcinogenesis of cancer cells [26, 27]. Recently, it has been reported that MrgprF (GPR168) acts as a tumor suppressor in cutaneous melanoma by restraining Akt signaling [28]. Therefore, the RT-qPCR and western blot were performed to determine the expression of some markers in these signaling pathways. The results showed that mRNA expression of *β-catenin*, *Myc*, *CyclinD1* and *CDK4* were significantly reduced in Akt signaling pathway (Fig 3F). The results of western blot showed similar results with the RT-qPCR findings (Figs 3G and 3H and S3–S5). Meanwhile, it was also found that the expression level of p-Akt and p-GSK-3β in pCDH-GPR168 was lower as compared to the negative control group, whereas the expression levels of Akt and GSK-3β had no significant difference compared to the control group (Figs 3G and 3H and S3–S5). In addition, the expression of CDK4, CyclinD1 and Myc was reduced, which affected the cell cycle and cell differentiation [29–33]. The results showed that GPR168 inhibited cell proliferation and migration by Akt signaling pathway. At last, we conclude a signaling pathway model for GPR168 function in B16-F10 cells (Fig 3I).

In order to validate the results, a rescue experiment was conducted (Fig 4A). Anti-GPR168 pAbs, performing as a nanobody to block the function of GPR168, was introduced into the medium of pCDH-GPR168 B16-F10 cell line for cell culture at concentration of 100 ng/ml.

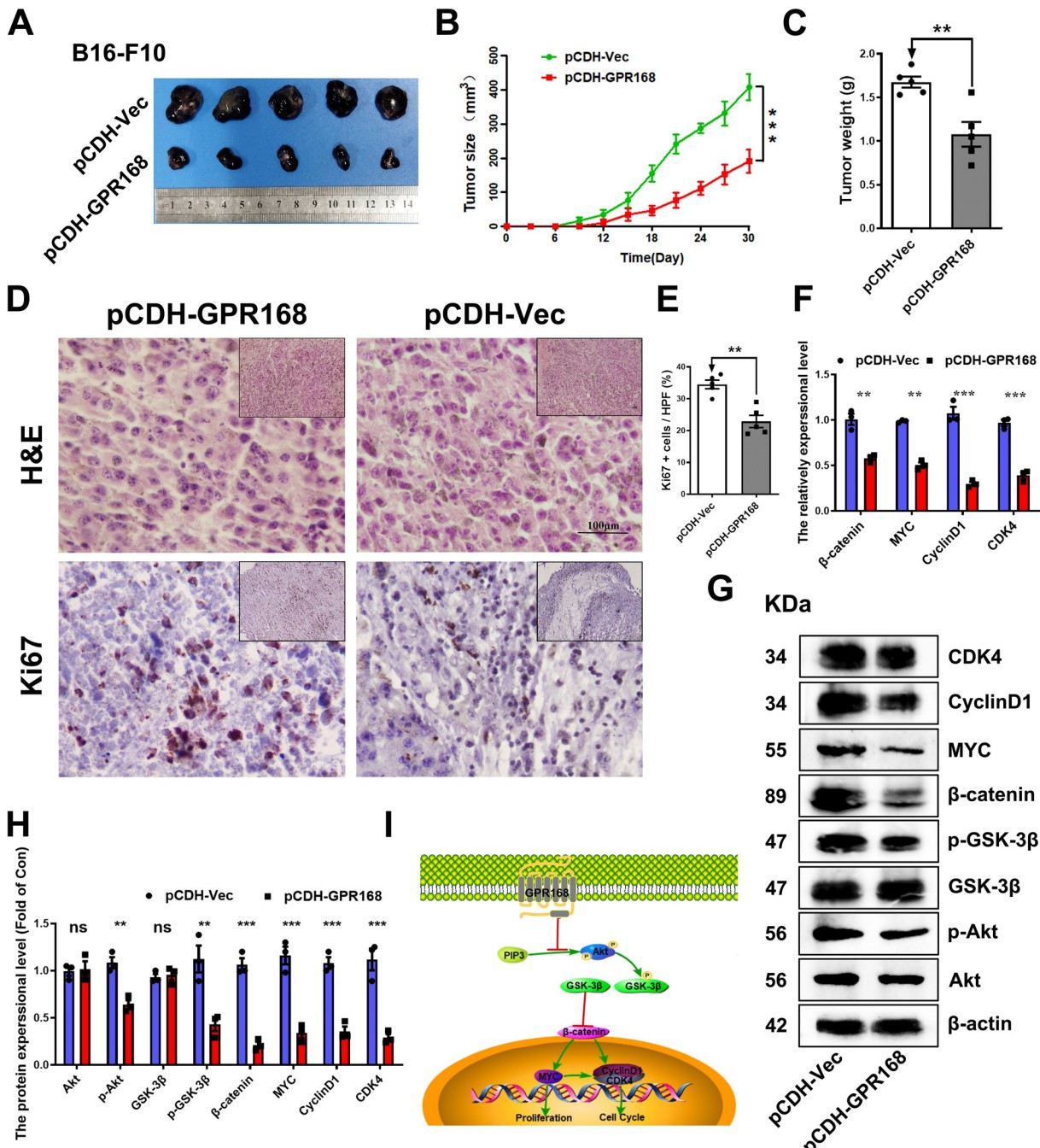

**Fig 3. Overexpression of GPR168 inhibited xenograft tumors growth *in vivo*.** (A) The sizes of allogenic tumors at day 30 were smaller by pCDH-GPR168, compared to those by pCDH-Vec. (B) GPR168 overexpression significantly reduced MM tumor growth in nude mice by tumor volume examination from 0 to 30 day. (C) Overexpression of GPR168 significantly suppressed the weights of allogeneic tumors. (D, E) the positive cells of Ki67 in allogenic tumor tissues. Scale bar is 100 μm, (E) Quantification for (D). (F) *β-catenin*, *Myc*, *CyclinD1*, *CDK4* mRNA expression in pCDH-GPR168 group and pCDH-Vec group cells (RT-qPCR). (G, H) Akt, p-Akt, GSK-3β, p- GSK-3β, β-catenin, Myc, CyclinD1, CDK4 protein expression in pCDH-GPR168 group and pCDH-Vec group (Western blotting). (I) A signaling pathway model for GPR168 function in B16-F10 cell line. β-actin as normalized protein. $^{*}P < 0.05$, $^{**}P < 0.01$, $^{***}P < 0.001$, *t*-test.

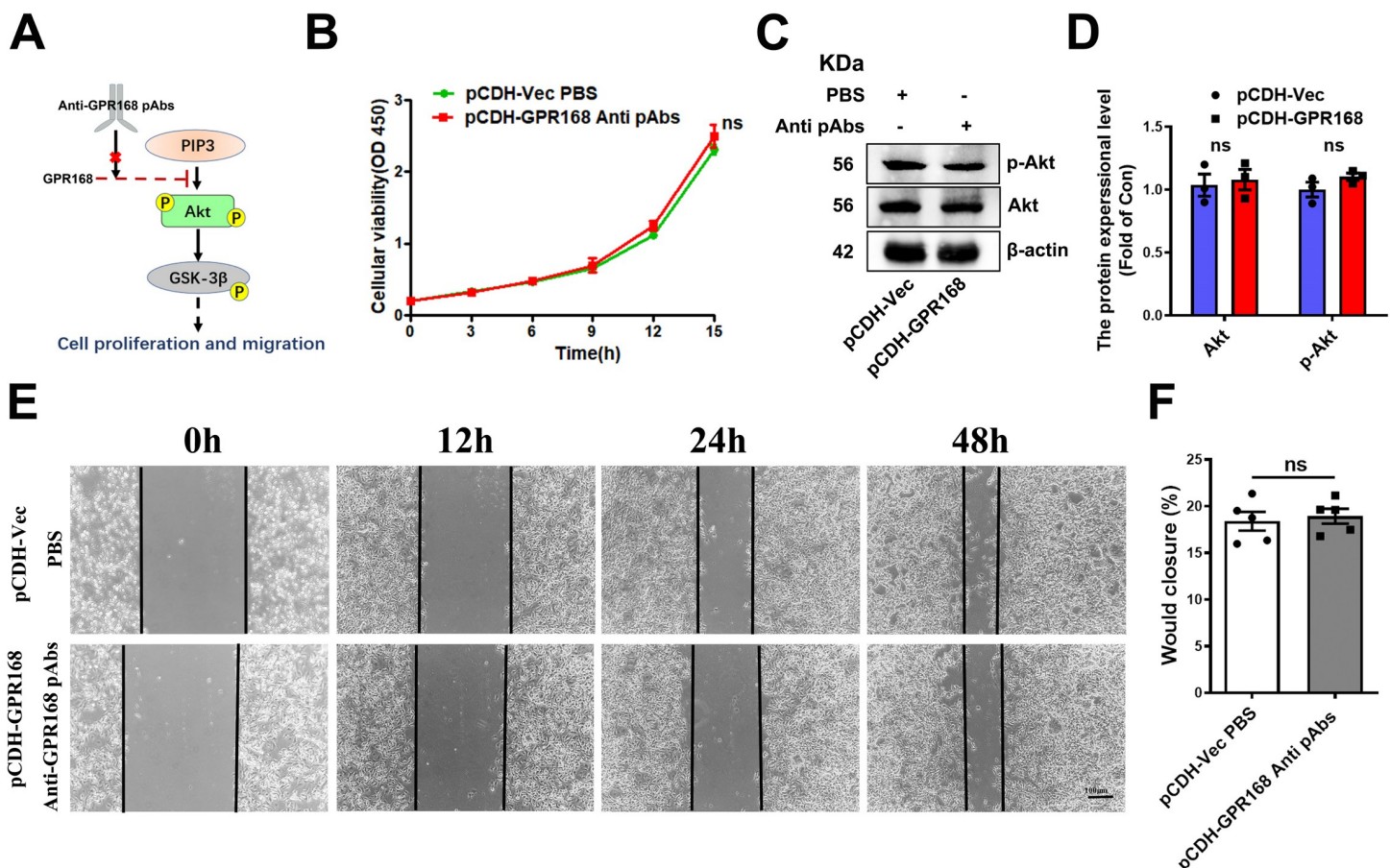

**Fig 4. A rescue experiment designed for GPR168.** (A) A working model designed for rescue experiment, the Anti-GPR168 pAbs can block the GPR168 function and resulted in the downstream repercussion. (B) CCK-8 assays after cultured in Anti-GPR168 pAbs medium (100 ng/ml), the result of two groups has no difference. (C, D) Western botting analysis of Akt and p-Akt protein expression in GPR168 overexpression B16-F10 melanoma cells cultured in 100 ng/ml Anti-GPR168 pAbs medium, pCDH-Vec was set as control. β-actin as normalized protein. (E, F) Wound healing assay results of pCDH-GPR168 group with PBS and pCDH-Vec group with VHH-NKG2A medium, showed that the migration of two groups had no significant difference. Scale bar: 100 μm. (F) Relative width change (wound closure) comparing the 48h width with 0 h starting width (%), quantification for (E). $^{*}P < 0.05$, $^{**}P < 0.01$, $^{***}P < 0.001$, $t$-test.

The control group was added with equal volume of PBS solution. After several generations of cell culture, cell pellets were harvested for further study. The results of western blotting showed that the expression of Akt and p-Akt in B16-F10 cells with pCDH-GPR168 had no significant difference compared to the control group (Figs 4C and 4D and S6). Furthermore, the CCK-8 assay showed that the proliferation of the experimental group and the control group had no significant difference (Fig 4B). The wound healing assay showed that the migration had no significant difference between two groups (Fig 4E and 4F). The results provide further support for the notion that GPR168 inhibits cell proliferation and migration by participating in the Akt signaling pathway.

## Discussion

Malignant melanoma (MM) is characterized by its high proliferation and propensity for metastasis. Due to the limited effective treatment methods available for patients with advanced melanoma, the mortality rates remain exceedingly high, significantly impacting the quality of life for patients. In recent years, there has been notable progress in the immunotherapy for MM.

Various combination of approaches, including immunotherapeutic agents and molecular inhibitors targeting single or multiple pathway(s) have been employed for clinical control of MM [6]. At the same time, clarifying the mechanism underlying MM pathology and progression is crucial for improving the methods of new combination therapies to promote tumor remission and prolong survival [34, 35]. In the study, GPR168 in B16-F10 cells and nude mouse melanoma model not only provides preliminary data to investigate melanoma, but also provides insight for the further treatment of human and animal melanoma. Although great progress has been made in the diagnosis and treatment of MM, potential novel biomarkers that may provide new insight into the prognostication of melanoma are still in urgent demand. Our group used comprehensive integrative bioinformatics methodologies to identify new biomarkers of tumor progression. Among the candidate tumor-associated novel biomarkers, GPR168, a member of the MAS-related GPR family, was found to be down regulated in multiple tumors, including melanoma. This led us to speculate that it might play a significant role in the development of MM, so we conducted the following studies to investigate its potential functions.

GPCRs are membrane embedded receptors. They implicate in regulating pivotal biological and pathological functions and have become valuable anti-cancer drug targets. In this study, we found that GPR168 overexpression inhibited tumor cell proliferation and migration *in vitro*, and nude mouse tumor growth *in vivo*, respectively. Importantly, Anti-GPR168 pAbs blocked the function of GPR168 and restored the abilities of B16-F10 cell proliferation and migration. These findings suggest that GPR168 acts as tumor suppressor in MM. Documented findings have shown that the Akt signaling pathway is frequently activated in MM. Our findings indicate that the absence of GPR168 in MM lead to constitutive activation of PI3K/Akt signaling [36]. When the Akt signaling is activated by hormone receptors, it usually leads to the increased phenomenon of cell proliferation and migration, while decreased phenomenon of cell apoptosis. In the study, we found that GPR168 inhibited the phosphorylation of Akt, resulting in the decreased of phosphorylation in GSK-3β downstream. The increased expression of GSK-3β could inhibit the activity of β-catenin. β-catenin cascades down to the nuclear proteins Myc, CycinD1 and CDK4. CDKs are key regulators downstream of Myc, which stimulate cell cycle transition from one phase to the next [37]. CyclinD1 and CDK4 are the checkpoints of the G1/S phase of the cell cycle [38, 39]. GPR168 overexpression in MM caused the changes of cell signaling pathways and affected the cell cycle. Taken together, our findings indicate that GPR168 inhibited the proliferation and migration in B16-F10 cell lines through the Akt signal pathway. However, human bioinformatics data used but only mouse data from one cell-line included and no human data included in this study. Additionally, the precise mechanism by how GPR168 preferentially binds and inhibits the PI3K complex remains unclear. In future, elucidating the crystal structure, protein binding components, gene interacting network, and the *in vitro* catalytic studies will be pivotal in deciphering the underlying mechanism. Collectively, our findings identify GPR168 as a novel tumor suppressor in mouse melanoma B16-F10 cells.

## Conclusion

MM is a highly malignant tumor, and its global incidence is progressively rising each year. Therefore, understanding the molecular mechanisms of melanoma is of great significance for its accurate diagnosis, prognosis and treatment. In the present study, we found that GPR168 expressed in mouse melanoma B16-F10 cell is lower compared to normal melanocytes cells. Based on these findings, we assume that GPR168 may function as cancer-suppressing gene. Indepth investigation revealed that overexpression of GPR168 in B16-F10 cells significantly inhibited the proliferation, migration of cells and xenograft tumors growth. Furthermore, our

studies also showed that GPR168 have negative relationship with Akt signaling pathway, whereas the Akt signal pathway is associated to cell proliferation and migration. In summary, our findings suggest that GPR168 inhibits the proliferation and migration of B16-F10 and xenograph tumor growth via Akt pathway.

## Supporting information

**S1 Fig. Three repetitions of western bloting in Fig 1D and the original image.** Western blotting results showed the protein expression of GPR168 in mice melanoma cell line B16-F10, compared to normal melanocytes cells (normalized to *ß-actin*).
(JPG)

**S2 Fig. Three repetitions of western bloting in Fig 2C and the original image.** The efficiency of pCDH-GPR168 (GPR168 overexpression) was verified by western blotting, compared to scramble pCDH-Vec.
(JPG)

**S3 Fig. Repetition 1 of western bloting in Fig 3G and the original image.** Akt, p-Akt, GSK-3β, p- GSK-3β, β-catenin, Myc, CyclinD1, CDK4 protein expression in pCDH-GPR168 group and pCDH-Vec group.
(JPG)

**S4 Fig. Repetition 2 of western bloting in Fig 3G and the original image.** Akt, p-Akt, GSK-3β, p- GSK-3β, β-catenin, Myc, CyclinD1, CDK4 protein expression in pCDH-GPR168 group and pCDH-Vec group.
(JPG)

**S5 Fig. Repetition 3 of western bloting in Fig 3G and the original image.** Akt, p-Akt, GSK-3β, p- GSK-3β, β-catenin, Myc, CyclinD1, CDK4 protein expression in pCDH-GPR168 group and pCDH-Vec group.
(JPG)

**S6 Fig. Three repetitions of western bloting in Fig 4C and the original image.** Western botting analysis of Akt and p-Akt protein expression in GPR168 overexpression B16-F10 melanoma cells cultured in 100 ng/ml Anti-GPR168 pAbs medium, pCDH-Vec was set as control. β-actin as normalized protein.
(JPG)

**S1 Data. Minimal data set (RT-qPCR): The original data of RT-qPCR in Figs 1F, 2A and 3F.**
(XLSX)

**S2 Data. Minimal data set (image): Repetitions and statistics of immunofluorescence assay, transwell assay, immunohistochemistry and wound healing assay in Figs 2D, 2H, 2J, 3D and 4E.**
(DOCX)

## Acknowledgments

The authors also acknowledge the Medical Experimental Center of Shanxi Bethune Hospital for providing the necessary equipment for this work. Thanks to Mureed Abbas for polishing and revising the language of the paper.

## Author Contributions

**Conceptualization:** Xiang Guo, Yiyan Zhao.

**Data curation:** Xiang Guo, Peirong Bai, Congfang Guo, Xuewei Liu.

**Formal analysis:** Kaiyi Zhu.

**Funding acquisition:** Xiang Guo.

**Validation:** Zongliang Guo.

**Writing – original draft:** Xiang Guo.

**Writing – review & editing:** Xiaoyan Li, Yiyan Zhao.

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
