## [Decision Letter · Decision Letter 0]

2 Nov 2023

PONE-D-23-27240GPR168 functions as a tumor suppressor in mouse melanoma by restraining Akt signaling pathwayPLOS ONE

Dear Dr. Zhao,

Thank you for submitting your manuscript to PLOS ONE. After careful consideration, we feel that it has merit but does not fully meet PLOS ONE’s publication criteria as it currently stands. Therefore, we invite you to submit a revised version of the manuscript that addresses the points raised during the review process.

We look forward to receiving your revised manuscript.

Kind regards,

Suzie Chen

Academic Editor

PLOS ONE

Journal Requirements:

- https://www.nature.com/articles/s41419-018-0299-1#citeas

In your revision ensure you cite all your sources (including your own works), and quote or rephrase any duplicated text outside the methods section. Further consideration is dependent on these concerns being addressed.

 "GX: the Fundamental Research Program of Shanxi Province (202203021222348), National Natural Science Foundation of China (8230089), and Scientific research start-up fund project for talent introduction of Shanxi Bethune Hospital (2023RC22). Xiang Guo designed the experiments, performed most of the experiments and wrote the paper."

7. PLOS ONE now requires that authors provide the original uncropped and unadjusted images underlying all blot or gel results reported in a submission’s figures or Supporting Information files. This policy and the journal’s other requirements for blot/gel reporting and figure preparation are described in detail at https://journals.plos.org/plosone/s/figures#loc-blot-and-gel-reporting-requirements and https://journals.plos.org/plosone/s/figures#loc-preparing-figures-from-image-files. When you submit your revised manuscript, please ensure that your figures adhere fully to these guidelines and provide the original underlying images for all blot or gel data reported in your submission. See the following link for instructions on providing the original image data: https://journals.plos.org/plosone/s/figures#loc-original-images-for-blots-and-gels. 

**Additional Editor Comments:**

Three reviewers have carefully review your manuscript and suggested major revisions. Please read each of their comments carefully and answer each concern point by point, if necessary please also provide supporting data.

Reviewers' comments:

Reviewer's Responses to Questions

**Comments to the Author**

1. Is the manuscript technically sound, and do the data support the conclusions?

Reviewer #1: Partly

Reviewer #2: No

Reviewer #3: Yes

2. Has the statistical analysis been performed appropriately and rigorously? 

Reviewer #1: Yes

Reviewer #2: I Don't Know

Reviewer #3: Yes

3. Have the authors made all data underlying the findings in their manuscript fully available?

Reviewer #1: Yes

Reviewer #2: Yes

Reviewer #3: Yes

4. Is the manuscript presented in an intelligible fashion and written in standard English?

Reviewer #1: Yes

Reviewer #2: No

Reviewer #3: Yes

5. Review Comments to the Author

Reviewer #1: In this current article titled "GPR168 functions as a tumor suppressor in mouse melanoma by restraining Akt

signaling pathway", authors reported the tumor suppressing properties of GPR168 in mouse melanoma. The paper contains its own merit considering the findings of a noble targets of melanoma however, it requires MAJOR REVISION before considering for publications.

(1) Its unrelated to show TCGA human data set while validate with murine cell lines. Author should consider incorporation of at least one human melanoma cell lines.

(2) The data extracted from GEPIA and cBioportal even unrealistically matched with current findings as TCGA melanoma cohort published in 2015 by pancancer atlas considered both primary and metastatic tumor for their study. This cohort doesn't purely represent the story related with malignant melanoma.

(3) What's the rationale of overexpression total AKT expression in pCDH-GPR168 cells?

(4) After blocking the function of GPR168, p-AKT expression didn't change and cell viability didn't alter compared with empty vector. How this target can be trusted to became a control factor of tumorigenesis?

(5) The figures look blurred.

Reviewer #2: Although the bioinformatics analysis shows a decreased expression of GPR168 across different human tumors including melanoma, the paper falls short of showing a causality of low expressed GPR168 on melanoma.The entire paper relied on one mouse melanoma cell line (B16-F10) with or without ectopic expression of GPR168. There are inconsistencies in the results even with this cell line. The authors mistakenly considered injecting mouse melanoma cell line F10 into nude mice an allogeneic model.

Reviewer #3: The manuscript “GPR168 functions as a tumor suppressor in mouse melanoma by restraining Akt signaling pathway” by Guo et al is very interesting as there are only a few tumor suppressors. Please see below my specific comments.

Specific Comments:

1. Expression of GPR in B10-F10 should be shown by immunoblotting. Otherwise, we assume that it is there without any proof.

2. CDK4 is down but no change in cyclin D1. Also, how then there is a decrease in Ki67?

Minor Comments: Figures are of poor quality especially IHC. Effort must be made to produce high quality figures.

6. PLOS authors have the option to publish the peer review history of their article (what does this mean?). If published, this will include your full peer review and any attached files.

Reviewer #1: No

Reviewer #2: No

Reviewer #3: **Yes: **Dayanidhi Raman

---

## [Author Response · Author response to Decision Letter 0]

11 Jan 2024

Responses to Reviewer 1

In this current article titled "GPR168 functions as a tumor suppressor in mouse melanoma by restraining Akt signaling pathway", authors reported the tumor suppressing properties of GPR168 in mouse melanoma. The paper contains its own merit considering the findings of a noble targets of melanoma however, it requires MAJOR REVISION before considering for publications.

Response: Thanks for reviewer’s comments. We have made MAJOR REVISION according to your suggestions.

1. Its unrelated to show TCGA human data set while validate with murine cell lines. Author should consider incorporation of at least one human melanoma cell lines.

Response: We applied Cross-Value Association Analysis (CVAA) to large-scale pan-cancer transcriptome data generated by The Cancer Genome Atlas (TCGA) [1] and found that GPR168 is one of Differentially Expressed Genes (DEGs). Subsequently, we found that only one paper reported that GPR168 (MrgprF) acts as a tumor suppressor in cutaneous melanoma by restraining PI3K/Akt signaling pathway, and this result was confirmed by four human melanoma cells [2]. In order to further verify whether GPR168 involves in the PI3K/Akt signaling pathway, mouse melanoma B16-F10 cells were used to perform cell experiments in vitro and xenograph tumor models in vivo.

2. The data extracted from GEPIA and cBioportal even unrealistically matched with current findings as TCGA melanoma cohort published in 2015 by pancancer atlas considered both primary and metastatic tumor for their study. This cohort doesn't purely represent the story related with malignant melanoma.

Response: Thanks for reviewer’s professional comments. As you said, this cohort doesn't purely represent the story related with malignant melanoma. However, we want to collect as much public data as possible to make a preliminary analysis of whether GPR168 is related to melanoma. Therefore, we used as many melanoma data from multiple databases as possible to make a preliminary statistical analysis of the correlation, making it more statistical and scientific to show that GPR168 is associated with melanoma. However, this result could not prove that GPR168 is directly related to melanoma, so we performed a series of follow-up experiments in vitro and in vivo (Fig 2-4).

3. What's the rationale of overexpression total AKT expression in pCDH-GPR168 cells?

Response: Thanks for reviewer’s question, which allows us to discover new problems. We repeated the western blotting experiments and found that the expression of AKT in pCDH-Vec and pCDH-GPR168 had no significant difference (Fig 3F).

Fig 3. Overexpression of GPR168 inhibited xenograft tumors growth in vivo. (A) The sizes of allogenic tumors at day 30 were smaller by pCDH-GPR168, compared to those by pCDH-Vec. (B) GPR168 overexpression significantly reduced MM tumor growth in nude mice by tumor volume examination from 0 to 30 day. (C) Overexpression of GPR168 significantly suppressed the weights of allogeneic tumors. (D, E) the positive cells of Ki67 in allogenic tumor tissues. Scale bar is 100 μm, (E) Quantification for (D). (F) Akt, p-Akt, GSK-3β, p- GSK-3β, β-catenin, Myc, CyclinD1, CDK4 protein expression in pCDH-GPR168 group and pCDH-Vec group (Western blotting). (G) β-catenin, Myc, CyclinD1, CDK4 mRNA expression in pCDH-GPR168 group and pCDH-Vec group cells (RT-qPCR). (H) A signaling pathway model for GPR168 function in B16-F10 cell line. β-actin as normalized protein. *P < 0.05, **P < 0.01, ***P < 0.001, t-test.

4. After blocking the function of GPR168, p-AKT expression didn't change and cell viability didn't alter compared with empty vector. How this target can be trusted to became a control factor of tumorigenesis?

Response: The overexpression of GPR168 could inhibit the proliferation and migration of melanoma, as well as the formation of xenograft tumors (Fig 2). After a series of functional experiments, it was found that GPR168 could function through the AKT signaling pathway (Fig 3). We preliminarily concluded that GPR168 is a tumor suppressor (melanoma) gene. Subsequently, we performed rescue experiment, and found that Anti-GPR168 pAbs could block the function of pCDH-GPR168 plasmid in B16-F10 tumor cells (Fig 4), which further confirmed that GPR168 could inhibit the tumorigenesis of melanoma through AKT signaling pathway. Taken together, we conclude that GPR168 could be a control factor of tumorigenesis (melanoma).

5. The figures look blurred.

Response: Thanks for reviewer’s comments. We repeated Western blotting experiment and re-made the Fig 1D, 2B, 3F and 4C.

Responses to Reviewer 2

Although the bioinformatics analysis shows a decreased expression of GPR168 across different human tumors including melanoma, the paper falls short of showing a causality of low expressed GPR168 on melanoma. The entire paper relied on one mouse melanoma cell line (B16-F10) with or without ectopic expression of GPR168. There are inconsistencies in the results even with this cell line. The authors mistakenly considered injecting mouse melanoma cell line F10 into nude mice an allogeneic model.

Response: Thanks for reviewer’s comments. In this paper, we found that the expression of GPR168 might be correlated with the tumorigenesis of melanoma (Fig 1). Subsequently, we found that one paper reported that GPR168 (MrgprF) acts as a tumor suppressor in cutaneous melanoma by restraining PI3K/Akt signaling pathway, and this result was confirmed by four human melanoma cells [2]. In order to further explore the function of GPR168 in melanoma cells, cell experiments in vitro (Fig 2) and xenograph tumor models in vivo (Fig 3) were performed and the results showed the overexpression of GPR168 could inhibit the tumorigenesis of melanoma and GPR168 involved in the PI3K/Akt signaling pathway. Furthermore, we confirmed this conclusion by rescue experiment (Fig 4). 

Responses to Reviewer 3

The manuscript “GPR168 functions as a tumor suppressor in mouse melanoma by restraining Akt signaling pathway” by Guo et al is very interesting as there are only a few tumor suppressors. Please see below my specific comments.

Specific Comments:

1. Expression of GPR in B10-F10 should be shown by immunoblotting. Otherwise, we assume that it is there without any proof.

Response: Thanks for reviewer’s comments. We have performed Western blotting as reviewer’s suggestion. The western blotting results showed that the B16-F10 cell line with stable GPR168 expression was successfully constructed (Fig 2B).

Fig 2. Overexpression of GPR168 inhibited B16-F10 cell proliferation and migration. (A, B) The efficiency of pCDH-GPR168 (GPR168 overexpression) was verified by RT-qPCR and western botting, compared to scramble pCDH-Vec. (C, D) GPR168 overexpression inhibited cell proliferation in B16-F10 cells by Ki67 Immunofluorescence assay. Scale bar is 100 μm. (E, F) Result of CCK-8 assay (E) and Growth Curve (F) showed that overexpression of GPR168 significantly inhibited B16-F10 cell proliferation. (G, H) overexpression of GPR168 decreased the migration ability of B16-F10 by transwell assay at OD570. Scale bar is 100 μm. (I, J) Wound healing assay showed that overexpression of GPR168 inhibited the migration of B16-F10 cells. Scale bar is 100 μm. *P < 0.05, **P < 0.01, ***P < 0.001, t-test.

2. CDK4 is down but no change in cyclin D1. Also, how then there is a decrease in Ki67?

Response: Thanks for reviewer’s question, which allows us to discover new problems. We repeated the western blotting experiments and found that both of CDK4 and cyclin D1 protein were decreased (Fig 3F).

Fig 3. Overexpression of GPR168 inhibited xenograft tumors growth in vivo. (A) The sizes of allogenic tumors at day 30 were smaller by pCDH-GPR168, compared to those by pCDH-Vec. (B) GPR168 overexpression significantly reduced MM tumor growth in nude mice by tumor volume examination from 0 to 30 day. (C) Overexpression of GPR168 significantly suppressed the weights of allogeneic tumors. (D, E) the positive cells of Ki67 in allogenic tumor tissues. Scale bar is 100 μm, (E) Quantification for (D). (F) Akt, p-Akt, GSK-3β, p- GSK-3β, β-catenin, Myc, CyclinD1, CDK4 protein expression in pCDH-GPR168 group and pCDH-Vec group (Western blotting). (G) β-catenin, Myc, CyclinD1, CDK4 mRNA expression in pCDH-GPR168 group and pCDH-Vec group cells (RT-PCR). (H) A signaling pathway model for GPR168 function in B16-F10 cell line. β-actin as normalized protein. *P < 0.05, **P < 0.01, ***P < 0.001, t-test.

3. Minor Comments: Figures are of poor quality especially IHC. Effort must be made to produce high quality figures.

Response: Sorry for poor quality of figures. We have made revise in Fig 3D.

References:

[1] Li QG, He YH, Wu H, Yang CP, Pu SY, Fan SQ, et al. A normalization-free and nonparametric method sharpens largescale transcriptome analysis and reveals common gene alteration patterns in cancers. Theranostics. 2017;7(11):2888-2899. https://doi.org/10.7150/thno.19425. eCollection 2017 PMID: 28824723

[2] Shen QS, Han YF, Wu K, He YM, Jiang XL, Liu PS, et al. MrgprF acts as a tumor suppressor in cutaneous melanoma by restraining PI3K/Akt signaling. Signal Transduct Target Ther. 2022;7(1):147. Epub 2022/05/04. https://doi.org/10.1038/s41392-022-00945-9 PMID: 35504869

---

## [Decision Letter · Decision Letter 1]

6 Mar 2024

PONE-D-23-27240R1GPR168 functions as a tumor suppressor in mouse melanoma by restraining Akt signaling pathwayPLOS ONE

Dear Dr. Zhao,

Thank you for submitting your manuscript to PLOS ONE. After careful consideration, we feel that it has merit but does not fully meet PLOS ONE’s publication criteria as it currently stands. Therefore, we invite you to submit a revised version of the manuscript that addresses the points raised during the review process.

Minor revision has been suggested by the reviewer based on your submitted revised manuscript.

We look forward to receiving your revised manuscript.

Kind regards,

Suzie Chen

Academic Editor

PLOS ONE

Journal Requirements:

Reviewers' comments:

Reviewer's Responses to Questions

**Comments to the Author**

1. If the authors have adequately addressed your comments raised in a previous round of review and you feel that this manuscript is now acceptable for publication, you may indicate that here to bypass the “Comments to the Author” section, enter your conflict of interest statement in the “Confidential to Editor” section, and submit your "Accept" recommendation.

Reviewer #4: (No Response)

2. Is the manuscript technically sound, and do the data support the conclusions?

Reviewer #4: Yes

3. Has the statistical analysis been performed appropriately and rigorously? 

Reviewer #4: Yes

4. Have the authors made all data underlying the findings in their manuscript fully available?

Reviewer #4: Yes

5. Is the manuscript presented in an intelligible fashion and written in standard English?

Reviewer #4: No

6. Review Comments to the Author

Reviewer #4: In this manuscript by Zhao and colleagues they describe the function of GPR168 in melanoma. Their mouse data suggests that GPR168 may act as a tumor suppressor in melanoma and they have shown data suggesting this but clarification and limitations of the study need to be discussed to improve the quality of the article.

Comments:

Data point distribution for all graphs should be included in each figure.

Showing representative western immunoblots is not enough, data needs to be normalized, graphed, and individual data points must be shown.

Limitations of this study must be stated in the discussion i.e human bioinformatics data used but only mouse data from one cell-line included and no human data included. Additionally, the limitations identified by the other reviewers must be stated.

The comment made by the authors,”We repeated the western blotting experiments and found that both of CDK4 and cyclin D1 protein were decreased (Fig 3F).” based on reviewer 3’s comments of the fact that cyclin D1 did not decrease this is quite concerning since it suggest reproducibility issues which is why the authors should not show just the representative western immunoblots but rather both representative western immunoblots and quantifications of the western immunoblots normalized.

7. PLOS authors have the option to publish the peer review history of their article (what does this mean?). If published, this will include your full peer review and any attached files.

Reviewer #4: **Yes: **Kevinn Eddy

---

## [Author Response · Author response to Decision Letter 1]

26 Mar 2024

1. Is the manuscript presented in an intelligible fashion and written in standard English?

Reviewer #4: No

Response: We feel sorry for our poor writings, however, we do invite a friend of us who is a native English speaker from the USA to help polish our article. And we hope the revised manuscript could be acceptable for you.

2. Data point distribution for all graphs should be included in each figure.

Response: Thanks for the comment. We have now included data point distribution in all graphs.

3. Showing representative western immunoblots is not enough, data needs to be normalized, graphed, and individual data points must be shown.

Response: Thanks for your suggestion. We have now normalized western blot data and showed in Fig 1E, Fig 2B, Fig 3H, Fig 4D.

4. Limitations of this study must be stated in the discussion i.e human bioinformatics data used but only mouse data from one cell-line included and no human data included. Additionally, the limitations identified by the other reviewers must be stated.

Response: Thanks for the comment. We have now pointed this limitation in discussion (Lines 359-360). 

5. The comment made by the authors, “We repeated the western blotting experiments and found that both of CDK4 and cyclin D1 protein were decreased (Fig 3F).” based on reviewer 3’s comments of the fact that cyclin D1 did not decrease this is quite concerning since it suggest reproducibility issues which is why the authors should not show just the representative western immunoblots but rather both representative western immunoblots and quantifications of the western immunoblots normalized.

Response: Thanks for your comments. We repeated three times each western blot. We have now showed all western blot results in Supporting information and quantifications of the western immunoblots normalized in Fig 1E, Fig 2B, Fig 3H, Fig 4D.

---

## [Decision Letter · Decision Letter 2]

28 Mar 2024

GPR168 functions as a tumor suppressor in mouse melanoma by restraining Akt signaling pathway

PONE-D-23-27240R2

Dear Dr. Zhao,

We’re pleased to inform you that your manuscript has been judged scientifically suitable for publication and will be formally accepted for publication once it meets all outstanding technical requirements.

Kind regards,

Suzie Chen

Academic Editor

PLOS ONE

Additional Editor Comments (optional):

Reviewers' comments:

Reviewer's Responses to Questions

**Comments to the Author**

1. If the authors have adequately addressed your comments raised in a previous round of review and you feel that this manuscript is now acceptable for publication, you may indicate that here to bypass the “Comments to the Author” section, enter your conflict of interest statement in the “Confidential to Editor” section, and submit your "Accept" recommendation.

Reviewer #4: All comments have been addressed

2. Is the manuscript technically sound, and do the data support the conclusions?

Reviewer #4: Yes

3. Has the statistical analysis been performed appropriately and rigorously? 

Reviewer #4: Yes

4. Have the authors made all data underlying the findings in their manuscript fully available?

Reviewer #4: Yes

5. Is the manuscript presented in an intelligible fashion and written in standard English?

Reviewer #4: Yes

6. Review Comments to the Author

Reviewer #4: All comments comments have been addressed and article should be accepted since all comments have been addressed however minor english improvements are required.

7. PLOS authors have the option to publish the peer review history of their article (what does this mean?). If published, this will include your full peer review and any attached files.

Reviewer #4: **Yes: **Kevinn Eddy

---

## [Editor Report · Acceptance letter]

15 May 2024

PONE-D-23-27240R2 

PLOS ONE

Dear Dr. Zhao, 

I'm pleased to inform you that your manuscript has been deemed suitable for publication in PLOS ONE. Congratulations! Your manuscript is now being handed over to our production team.

Kind regards, 

on behalf of

Dr. Suzie Chen 

Academic Editor

PLOS ONE